# Current Clinical Paradigm and Therapeutic Advancements in Thymic Malignancies: A Narrative Review

**DOI:** 10.3390/cancers17223622

**Published:** 2025-11-11

**Authors:** Douglas Dias e Silva, Beatriz Viesser Miyamura, Isa Mambetsariev, Jeremy Fricke, Javier Arias-Romero, Amit A. Kulkarni, Ajaz Khan, Debora S. Bruno, Jyoti Malhotra, Abigail Fong, Jae Kim, Colton Ladbury, Arya Amini, Gustavo Schvartsman, Ravi Salgia

**Affiliations:** 1Dayan-Daycoval Family Hematology and Oncology Center, Hospital Israelita Albert Einstein, São Paulo 05652-900, SP, Brazil; 2Department of Medical Oncology and Therapeutics Research, City of Hope National Medical Center, Duarte, CA 91010, USAjfricke@coh.org (J.F.);; 3Department of Medical Oncology and Therapeutics Research, City of Hope Phoenix, Goodyear, AZ 85338, USA; 4Department of Medical Oncology and Therapeutics Research, City of Hope Chicago, Chicago, IL 60611, USA; 5Department of Medical Oncology and Therapeutics Research, City of Hope Atlanta, Atlanta, GA 30265, USA; 6Department of Surgery, City of Hope, Duarte, CA 91010, USA; 7Department of Radiation Oncology, City of Hope, Duarte, CA 91010, USA

**Keywords:** thymic epithelial tumors, targeted therapy, immunotherapy, precision medicine

## Abstract

Thymic malignancies are a rare diverse group of tumors that occur due to dysfunction in thymic cells and are known to have a poor prognosis. The rare nature of this disease and the lack of mutations that can be targeted with inhibitors in this disease subtype have made it difficult to develop new effective drugs. However, more recently, novel classes of drugs such as immunotherapy and tyrosine kinase inhibitors have shown promising results in clinical trials and more drugs are being developed based on novel preclinical findings. Therefore, in this study we review the clinical characteristics of this disease, evaluate the mutations involved, explain the current standard of care, explore drug resistance, and describe in detail the ongoing novel drug results from in-human clinical trials.

## 1. Introduction

Thymic malignancies comprise a diverse group of rare tumors, with thymic epithelial tumors (TETs) being the most common subtype [1]. TETs are divided into two main groups: thymomas and thymic carcinomas, which are characterized by epithelial cell morphology, preservation of organ-specific characteristics, presence of mixed immature T cells, and degree of pleomorphism and nuclear atypia [2]. Due to the role of the thymus in adaptive immunity, where T cell maturation occurs, it is common to observe an association with autoimmune events in clinical practice [3]. A significant proportion of patients (approximately 30–44%) develop various autoimmune disorders, such as myasthenia gravis [3,4]. Conversely, individuals with thymic carcinoma rarely exhibit autoantibody-induced diseases [4]. Thymoma classification systems have limited effectiveness in precisely predicting prognosis and disease progression; therefore, staging remains the sole method for predicting clinical behavior [5,6]. The optimal approach for treating early-stage thymic epithelial tumors is surgery, which involves removing the tumor and adjacent tissues [7]. Surgery can be minimally invasive or open, depending on the patient’s specific characteristics [8]. The rarity of this tumor has long prevented its inclusion in extensive phase II and III clinical trials. Consequently, complementary and subsequent lines of treatment beyond surgery and radiation therapy, chemotherapy was initially the sole option, and the development of new drugs has progressed slowly. However, in the past decade, innovations in the treatment of advanced stages have emerged, particularly immunotherapy, tyrosine kinase inhibitors, and Hyperthermic Intrathoracic Chemotherapy (HITHOC) [9]. In this article, we review the histology and molecular biology of thymic neoplasms and discuss the latest treatment trends, current standard-of-care regimens, and research in progress on novel therapies.

## 2. Incidence and Epidemiology

TETs are rare compared to other thoracic neoplasms, with an incidence rate of 1.3–2.68 per million inhabitants per year in the United States [10,11]. Nevertheless, they are responsible for up to 50–63% of anterior mediastinal masses resected in adults above 20 years old [11]. When including non-resected radiologic series, this number is smaller, around 5.6% of all solitary mediastinal lesions, because of lesions that are not operated [12]. In this group, the diagnosis of thymoma is far more common than that of thymic carcinoma. However, long-term data may be inconsistent due to classification challenges, as it was only in 1999 when the second edition of the World Health Organization Classification of Tumors officially recognized thymomas and thymic carcinomas as distinct categories [13]. Assessing survival data is even more challenging, as a standardized system for thymomas was only introduced in the most recent edition of the Union for International Cancer Control (UICC) classification. These tumors are uncommon in individuals younger than 20 years, with the highest incidence in the age range of 44–74 years [11,14]. In the US, the distribution between female and male patients appears equal, but the incidence of thymoma is notably higher among Asians and Pacific Islanders. Ethnic differences could indicate a potential genetic influence, as these patients are usually younger and have higher incidence rates. The risk of secondary malignancies has also been studied, with conflicting data from different trials, possibly due to the sample size [15]. In a broad US cancer registry, patients with thymoma had a higher risk of developing B-cell non-Hodgkin’s lymphoma, allegedly due to abnormal functioning T-cells that induce or fail to control B-cell proliferation, with limited data suggesting an association with gastrointestinal tumors and soft tissue sarcomas [10]. It has been established that thymomas are associated with autoimmune diseases, more commonly with myasthenia gravis (24.5–38.8%), but also with systemic lupus erythematosus (2.3–2.4%) and red cell aplasia (1.2–2.3%) [3,16]. The reason autoimmune disorders and paraneoplastic syndromes appear to be associated with thymic malignances may be explained by the primary function of thymic tissue. The maturation and differentiation of precursor cells into T cells is the final process in the negative selection and eradication of T cells, which develop specific receptors that target self-antigens. This step is important for avoiding autoimmunity and depends on two important factors: the action of autoimmune regulator genes and major histocompatibility (MHC) class II, along with other transcription factors. One of the proposed mechanisms for autoimmune diseases is deprivation or deficit in the expression of autoimmune regulator genes and MHC class II. Another mechanism could be the alteration of the microenvironment caused by thymic malignant cells that leads to immature autoreactive T cells bypassing thymic medulla elimination and gaining systemic circulation. In contrast, paraneoplastic syndromes may be triggered by thymoma cells and their ability to release endocrine or paracrine messengers, such as hormones, cytokines, or peptides, and induce systemic effects. Furthermore, these immune changes may increase the risk of infection in immunodeficient patients and various types of malignancies [17].

## 3. Histological Classification

In the 1960s thymomas were classified as invasive or non-invasive [18]. Bernatz et al. introduced a histological classification system that identified four subtypes: predominantly lymphocytic, predominantly epithelial, predominantly mixed, and predominantly spindle cells. The first three types were characterized by round epithelial cells and lymphocytes, with their ratios determining the subtype classification. Furthermore, they discovered that the predominant epithelial subtype is the most common type of invasive thymoma. Nevertheless, no prognostic information was implied by this classification [19]. In 1978, Levine et al. differentiated thymomas into noninvasive, benign, invasive, or malignant, with the latter being further classified into Type I (no or minimal atypia) and Type II (moderate or marked atypia), also known as thymic carcinoma. This classification also has a clinical aspect that considers both the invasion and degree of atypia [20]. It should be highlighted that the initial terminology ‘benign thymoma’ is no longer used, acknowledging that all thymic epithelial tumors carry some malignant potential [2]. In 2004 the WHO classification introduced a new histological categorization emphasizing the morphology of thymic epithelial cells (spindle-shaped or epithelioid), the extent of the lymphocytic component, and epithelial atypia (present or absent) [2].

This classification system focuses on the morphology of thymic epithelial cells, degree of lymphocytic infiltration, and occurrence or lack thereof of epithelial atypia. Type A, AB, and B1 thymomas have very good overall survival (OS) rates, with only 5–10% of mortality rates in 10 years. In contrast, the five-year survival rates for thymomas B2, B3, and C are approximately 75–84.9%, 70–84%, and 48%, respectively [21,22,23]. Thymomas rarely metastasize, whereas thymic carcinomas are more aggressive and often present with symptomatic extensive local invasion and distant metastases in the liver, lymph nodes, or bones [23]. Moran et al. suggested a simplified WHO classification system, categorizing lesions as well differentiated, poorly differentiated and atypical thymomas. The first is a typical thymoma that lacks cytological features of malignancy, also known as medullary thymoma. The second category is traditional thymic carcinomas that present numerous mitotic figures and other malignant features. The third category represents lesions with intermediate characteristics, also known as well-differentiated thymic carcinoma, featuring organotypic aspects of thymomas, but with areas of atypia and occasional mitoses [24]. The fifth edition of the World Health Organization (WHO) classification of Thymic Epithelial Tumors is presented in Table 1.

The classification systems extend beyond histopathological categorization, providing valuable guidance for physicians in routine clinical decision-making [25]. Patients with low-risk subtypes (A, AB, B1) usually undergo complete resection alone, as their indolent course and excellent long-term survival rarely justify adjuvant therapy (Appendix A) [8]. High-risk thymomas (B2, B3) often require extended resection with adjuvant radiotherapy, whereas thymic carcinoma demands multimodal treatment including surgery, radiotherapy, and chemotherapy [21].

**Table 1 cancers-17-03622-t001:** WHO Classification of Thymic Epithelial Tumors [26].

Subtypes				
Micronodular thymoma with lymphoid stroma	Lymphoepithelial carcinoma	Adenosquamuos carcinoma	Carcinosarcoma	Combined small cell carcinoma
Metaplastic thymoma	Adenocarcinoma, NOS	NUT carcinoma	Carcinoma, undifferentiated, NOS	Large cell neuroendocrine carcinoma
Lipofibroadenoma	Low-grade papillary adenocarcinoma	Mucoepidermoid carcinoma	Carcinoid tumor, NOS/neuroendocrine tumor, NOS	
Squamous cell carcinoma, NOS	Thymic carcinoma with adenoid cystic carcinoma-like features	Clear cell carcinoma	Typical carcinoid/neuroendocrine tumor, grade 1/2	
Basaloid carcinoma	Adenocarcinoma, enteric type	Sarcomatoid carcinoma	Small cell carcinoma	

## 4. Staging

Several staging systems exist for thymic malignancies [27]. In 1981, Masaoka et al. defined a prognostic classification system that was later complemented by Koga in 1994 [5,28]. This classification was subsequently expanded by the International Thymic Malignancy Interest Group (ITMIG). The most important features observed were integrity of the thymic capsule, invasion of the surrounding tissue and organs, pleural or pericardium dissemination, and metastasis [29]. The American Joint Committee on Cancer (AJCC) and Union for International Cancer Control (UICC) proposed the staging of thymomas and thymic carcinomas using the TNM System. The eighth edition of the TNM system aimed to create a unified framework for comparing the results across various studies. In this edition, survival data from more than 10,000 patients were analyzed to determine the staging criteria [30]. However, the treatment of these patients was determined using the Masaoka staging system [23]. Tumor staging continues to be the most relevant prognostic factor in thymomas, but both stage and histological subtypes need to be considered when predicting survival outcomes [31]. Since January 2025 the 9th Edition of the TNM classification for thymic epithelial tumors has been updated (Table 2), adding the inclusion of tumor size (cutoff of 5 cm), and a redefinition of T2 and T3 categories [32].

## 5. Genomic Landscape

Molecular analyses have shown that various histological subtypes of thymic epithelial tumors are associated with distinct mutations and molecular pathways [33]. Among adult tumors, thymomas have one of the lowest rates of somatic mutations, although copy-number variation tends to increase from type A thymomas to thymic carcinomas [34]. Mutations in GTF2I have been detected in up to 39% of thymic epithelial tumors (TETs), with a particularly high prevalence in type A thymomas (100%) and type AB thymomas (70%) [34]. These mutations, consistently occurring at codon L424H, suggest oncogenic activity because they are associated with increased expression of genes involved in alteration of cellular signaling, dysregulation of the cell cycle, and impact on immune response [35,36]. Thymic epithelial tumors generally have low tumor mutational burden (TMB), averaging 0.48 mutations per megabase. Thymic carcinomas exhibit a higher TMB than thymomas. Radovich et al. found that most thymic epithelial tumor cases showed minimal somatic copy number alterations, with large-scale CNAs occurring mainly in infrequently mutated genes [33,37,38]. c-KIT, a transmembrane growth factor with tyrosine kinase activity, is present in several cell types and is crucial for cell proliferation, survival, and differentiation [39]. Overexpression of c-KIT is observed in 50–88% of thymic carcinomas but is rare in thymomas (2–5%) and is linked to poor prognosis [40,41]. EGFR, a member of the ErbB family (HER1), is overexpressed in 33–100% of thymomas and 33–83% of thymic carcinomas [42,43,44]. Its expression correlates with tumor size, clinical stage, and histopathological classification, although EGFR amplification does not strongly correlate with its overexpression [45,46]. HER2 (Human Epidermal Growth Factor Receptor 2), also a member of the ErB family, is a target in various epithelial tumors but is rarely expressed in thymomas. One study reported that only a few thymomas (types A, B2, and B3) showed focal membranous staining, whereas thymic carcinomas more frequently exhibited strong membranous expression without gene amplification [46,47]. Aggressive subtypes and advanced stages of thymic epithelial tumors appear to have PI3K-mTOR mutations that have been identified in thymic carcinoma and AB thymoma [48]. IGF-1R overexpression is more common in thymic carcinomas (70%) than in thymomas (21%) and is linked to more aggressive histological subtypes and advanced disease stages [49,50]. Chromosomal abnormalities are present in all thymic epithelial tumor subtypes, including translocations, such as t (15;19), and deletions, such as 6p22-p25. In thymomas, common genetic changes include losses in chromosome regions 6p21.3 and 6q25.2 to 25.3 [51]. High programmed cell death ligand 1 (PD-L1) expression is linked to improved survival, although studies have offered mixed prognostic data [52,53]. PD-L1 expression ranges from 23% to 92% in thymomas and from 36% to 54% in thymic carcinomas [52,54]. Among adult solid malignant tumors, thymic epithelial tumors exhibit the lowest tumor mutational burden (TMB), and microsatellite instability is rare [55]. The key molecular pathways involved in the pathogenesis of thymic epithelial tumors are shown in Figure 1.

Another entity that has become more frequently diagnosed and can arise from the thoracic midline is NUT carcinoma, which is defined by the presence of NUTM1 gene rearrangements. This is a rare carcinoma that arises along midline structures such as the head, neck, or lungs. The most frequent fusion occurs between NUT and BRD4, but also with BRD3, NSD3, or others. This gene fusion leads to the formation of abnormally activated NUT proteins, which causes aberrant squamous cell growth. These carcinomas are frequently diagnosed at an advanced stage and have poor prognosis because of their aggressive behavior [56].

Early studies had an overall low frequency of genomic alterations; however, because this is a rare entity, there could be bias in the small sample size. Recently, retrospective studies using larger databases have reported a larger number of genomic alterations that could be used for the development of new targeted drugs. K. Kurokawa et al. published in 2023 real-world data from 794 patients from the Foundation Medicine Inc. (FMI) in the United States and the Center for Cancer Genomics and Advanced Therapeutics (C-CAT) in Japan [57]. In thymic carcinoma, the most common alterations were CDKN2A, TP53, and CDKN2B, with frequency of 39.9%, 30.2% and 24.6%, respectively; TMB higher than 10 mutations/Mb was present in 7.0% and MSI in 2.3%. In thymoma, the most common alterations were TP53, DNMT3A, and CDKN2A, with frequencies of 7.8%, 6.8%, and 5.8%, respectively; TMB higher than 10 mutations/Mb was present in 1.6% and MSI in 0.3% [57]. Similar results have been observed in a Japanese database. Another cohort of 174 patients with thymic carcinoma demonstrated that they had a higher frequency of alterations at 4.0/tumor, and 6% had a high TMB. CDKN2A, KIT, and PTEN/PI3K/MTOR pathways had genomic alterations, while 90 patients with thymoma demonstrated a low frequency of genomic alterations, with an average of 1.8/tumor and low TMB; CDKN2A/B and TP53 gene alterations were found in 10% of cases, and one patient had amplification of the NTRK1 gene. This reinforces the importance of genomic testing for potential precision medicine treatment options using targeted agents [12].

## 6. Current Systemic Therapy

The current first-line treatment for advanced thymic epithelial tumors is platinum-based chemotherapy, which has been established in small phase II studies [58]. The combination of cisplatin and etoposide, PAC (cisplatin, doxorubicin, and cyclophosphamide), or ADOC (cisplatin, doxorubicin, cyclophosphamide, and vincristine) showed response rates varying from 50% to more than 90% [59,60,61]. Carboplatin plus paclitaxel is also an option, but with lower response rates [62], with a recent study showing that the addition of ramucirumab to the combination had an OS of 43.8 months with a 57.6% ORR [63]. Following progression from first-line therapy, chemotherapy options for second-line treatment include single-agent pemetrexed [64], ifosfamide [65], and capecitabine plus gemcitabine [66]. In the neoadjuvant setting, a few retrospective studies showed no difference between upfront approach versus neoadjuvant treatment in regard to OS, PFS, pathological complete response or R0 resections [32,67].

## 7. HITHOC

For relapses in the pleura and pericardium, hyperthermic intrathoracic chemotherapy (HITHOC) as a complementary treatment to cytoreductive surgery has demonstrated efficacy and safety, particularly for malignant mesothelioma [68]. Some retrospective studies have evaluated the use of HITHOC for thymic neoplasms, with a small sample size but better outcomes, local control, benefits in disease-free interval, and acceptable morbidity [69,70,71,72,73]. There is an ongoing open, single-arm, prospective trial to study the safety and efficacy of cytoreductive surgery combined with hyperthermic intrathoracic chemotherapy (S-HITHOC) for treating thymic epithelial tumors with pleural spread or recurrence, the results of which are long-awaited [74].

## 8. Radiotherapy

Radiotherapy constitutes an important treatment of thymic epithelial tumors, but its indication is highly dependent on disease stage, type of surgical resection, and histological subtype [75]. In early-stage disease (stage I–IIA)**,** particularly in low-risk thymomas (types A to B1) with complete resection, adjuvant radiotherapy is not routinely recommended, because by reason of low recurrence rates and long-term survival is excellent after surgery alone [76,77,78].

In intermediate stages (stage IIB–III)**,** especially in B2 and B3 thymomas with capsular invasion, adjuvant radiotherapy is commonly indicated to improve local control based on retrospective analyses showing reduced recurrence in patients receiving radiotherapy. Nevertheless, studies based on large global, regional, and national databases have reported conflicting results [79], In advanced or high-risk disease (stage IVA–IVB and thymic carcinoma), the role of radiotherapy is more consistent. Postoperative radiation is recommended for R1 or R2 resections, and for thymic carcinoma regardless of stage, given its aggressive biology. When surgery is not feasible, definitive radiotherapy, frequently combined with platinum-based chemotherapy, represents the standard local treatment approach [77,80].

Currently multiple techniques are indicated on the treatment of thymic malignancies. Modern approaches such as intensity-modulated radiotherapy (IMRT), volumetric modulated arc therapy (VMAT), and image-guided radiotherapy (IGRT) allow dose escalation with better sparing of adjacent critical structures [80]. In cases of pleural dissemination, hemithoracic IMRT and proton beam therapy have shown encouraging preliminary results, with acceptable safety profiles and improved local control [81]. Another option is 4D imaging, when available, which accounts for tumor or postoperative bed motion to delineate an appropriate target [82].

## 9. Targeted Therapy Developments

The development of therapies that target specific genomic alterations has improved the treatment of numerous cancer types but remains a challenge in thymic malignancies due to the scarcity of actionable targets (Table 3) [12]. The multikinase inhibitors Lenvatinib, Sunitinib, Sorafenib, and the rapamycin pathway inhibitor, Everolimus, are options in later lines frequently for thymic carcinoma, even without known targets [83].

### 9.1. Anti-Angiogenic Agents

Lenvatinib is a multi-target tyrosine kinase inhibitor that targets VEGFR, FGFR KIT, and other kinases [84]. The efficacy of this drug for thymic carcinoma was evaluated in a phase 2 trial, REMORA. In this study involving 42 patients with advanced thymic carcinoma, 38.1% achieved partial response (PR), while 57.1% maintained stable disease (SD). The mPFS was 9.3 months, indicating some of the most promising outcomes for patients with metastatic thymic carcinoma. Lenvatinib is associated with a high risk of side effects and frequent dose reductions may be required [85]. There is an active, non-recruiting phase II study of lenvatinib combined with pembrolizumab for pretreated advanced B3-thymoma and thymic carcinoma that showed one-year OS of 85% [86]. Regorafenib inhibits various angiogenic and stromal receptor tyrosine kinases, including VEGFR1-3, tyrosine kinases with immunoglobulin-like and EGF-like domains 2, and PDGFRB. In one study, SD was observed in 85.7% of the patients (6 of 7), while 14.3% (1 patient) had PD [87]. Sunitinib, another multi-kinase inhibitor, also showed promising results, especially for the thymic carcinoma population and after the first-line treatment, as published in the STYLE trial [88]. Bevacizumab and ramucirumab did not show much relevant results, but combination therapy is currently under investigation [89].

### 9.2. KIT Inhibitors

KIT mutations are uncommon in thymic carcinoma; however, retrospective studies have shown that KIT-mutated thymic carcinoma has a meaningful clinical response to imatinib, although the association between KIT expression and response to Sunitinib and Sorafenib is uncertain [83,90]. A phase 2 trial investigating sunitinib in patients with thymic carcinoma reported that 3.6% achieved complete remission (CR), 17.9% had PR, and 67.9% exhibited SD) [88]. Retrospective data assessing the effectiveness of sorafenib in five patients with metastatic thymic carcinoma revealed that two patients (40%) experienced PR and another two patients (40%) maintained SD [91].

### 9.3. PI3K/mTOR Inhibitors

Another potential target for treatment is rapamycin (mTOR) with everolimus. This was tested in a phase II trial in which everolimus demonstrated a DCR of 93.8% with an mPFS of 16.6 months in 32 patients with cisplatin-pretreated thymoma and a DCR of 61.1% with a median PFS of 5.6 months in 18 patients with cisplatin-pretreated thymic carcinoma. Although off-label, everolimus could be an option for refractory cases of both thymomas and thymic carcinomas [92].

### 9.4. IGF1R Inhibitors

Cixutumumab is an IGF1R (Insulin-like Growth Factor 1 Receptor) inhibitor, which is a cell surface receptor that belongs to the tyrosine kinase receptor family and plays a key role in cell growth, survival, and development. IGF-1R is often overexpressed in various types of cancer. This drug was tested in a phase II multicenter open-label study on recurrent or refractory advanced thymic epithelial tumors. In the thymoma cohort, 5 (14%) of the 37 patients achieved PR, 28 had SD, and 4 had PD. In the thymic carcinoma cohort, zero of 12 patients achieved PR, 5 had SD, and 7 had PD. High toxicity and limited effectiveness of single-agent treatment have curbed the enthusiasm for the development of IGF-1R inhibitors, and there are currently no ongoing trials testing it for thymic epithelial tumors [93].

### 9.5. EGFR Inhibitors

EGFR overexpression is common in thymic epithelial tumors and associated with poor PFS and OS. Phase II trials of gefitinib or a combination of erlotinib and bevacizumab have reported poor efficacy [94].

### 9.6. Somatostatin Analogs

Somatostatin receptor 2 (SSTR2) is expressed in some thymic epithelial tumors, particularly thymomas. A study of 80 patients reported SSTR2 expression in 36.3% of thymic epithelial tumor cases, which had a direct relationship with younger age and reduced recurrence/metastasis-free survival. 68Ga-DOTATATE PET scans generally correlate with SSTR2 expression, suggesting the potential for guiding treatment with somatostatin analogs; however, larger studies are needed for validation [95]. Treatment with octreotide could be beneficial for patients with thymoma who have a positive octreotide scan, dotatate PET/CT positivity, or symptoms of carcinoid syndrome, but not for those with thymic carcinoma. In a trial of 42 patients with advanced thymoma or thymic carcinoma, 38 were fully assessed. Only patients with thymomas responded to treatment, with a survival rate of 86.6% after one year and 75.7% after two years [96].

### 9.7. Cyclin-Dependent Kinases Inhibitors

Deletion of the CDKN2A gene, which encodes CDK2, appears to be the most common alteration in thymic epithelial tumors and is associated with poor prognosis. An oral CDK inhibitor, milciclib, was tested in two phase II trials; the ORR was less than 5% in both trials, but the DCR was 75.9 and 83.3%, respectively. The first study reported an mPFS of 6.83 months and mOS of 24.18. The second trial had an mPFS of 9.76 months, and OS was not achieved [95]. Palbociclib, another CDK4/6 inhibitor, was tested in 48 patients with advanced thymic epithelial tumors after chemotherapy. Six of 48 patients (12.5%) achieved PR and the mOS was 26.4 months [94].

**Table 3 cancers-17-03622-t003:** Target therapy for recurrent thymic epithelial tumor.

Reference	Drug	Mechanism of Action	Key Clinical Evidence
Sato et al. [83]	Lenvatinib	Multi-target TKI (VEGFR, FGFR, KIT, PDGFR)	REMORA trial: PR 38.1%, SD 57.1%, mPFS 9.3 mo.
Agrafiotis et al. [85]	Regorafenib	Multi-target TKI (VEGFR1-3, PDGFRB, etc.)	Small study: SD in 85.7% (6/7 pts).
Proto et al. [86]	Sunitinib	Multi-target TKI (VEGFR, PDGFR, KIT)	STYLE trial: CR 3.6%, PR 17.9%, SD 67.9%.
Pagano et al. [89]	Sorafenib	Multi-target TKI (VEGFR, RAF, KIT, PDGFR)	5 pts: PR in 40%, SD in 40%.
Buti et al. [88]	Imatinib	KIT inhibition	Responses in KIT-mutated thymic carcinoma.
Arunachalam et al. [90]	Everolimus	mTOR inhibition	Phase II: Thymoma—DCR 93.8%, mPFS 16.6 mo; Carcinoma—DCR 61.1%, mPFS 5.6 mo.
Rajan et al. [91]	Cixutumumab	IGF-1R inhibition	Phase II: Thymoma—PR 14%, majority SD; Carcinoma—no PR, high toxicity.
Dapergola et al. [92]	Gefitinib/Erlotinib ± Bevacizumab	EGFR inhibition	Phase II: poor efficacy in TETs.
Roden et al. [93]; Loehrer et al. [94]	Octreotide	Somatostatin receptor (SSTR2) binding	Responses in thymomas only; 1-year OS 86.6%.
Roden et al. [93]	Milciclib	Pan-CDK inhibitor	ORR < 5%, but DCR 75–83%; mPFS ~7–10 mo.
Dapergola et al. [92]	Palbociclib	CDK4/6 inhibitor	48 pts: PR 12.5%, mOS 26.4 mo.

### 9.8. Emerging Targeted Therapies Currently Being Evaluated in Clinical Trials

Mesothelin is a cell surface protein whose high expression has been observed in thymic carcinoma, as well as in other malignancies, such as mesothelioma, ovarian cancer, and gastrointestinal tumors [97]. In vitro and in vivo models have suggested that a new antibody-drug-conjugate named anetumab ravtansine that targets mesothelin could be a potential targeted therapy for the treatment of thymic carcinoma [98]. Clinical data are expected, and the drug is currently being studied in Phase I/II trials for mesothelioma and ovarian cancer [99]. Other potential targets that require further investigation include histone deacetylases (HDAC) and Exportin-1 (XPO-1). Other clinical trials involving targeted therapies are summarized in Table 4.

## 10. Novel Immunotherapy Approaches

Despite the low TMB compared to other adult cancers, the high expression of PD-L1 and the presence of numerous CD8+ lymphocytes offer a compelling reason for treatment with immune checkpoint inhibitors (ICIs) [55,107]. Nevertheless, Thymoma and thymic carcinoma diverge in tumor biology, baseline autoimmunity, and the benefit–risk calculus of PD-1/PD-L1 blockade. Clinically, thymoma is disproportionately complicated by neuromuscular and cardiac immune-related adverse events (irAEs), including myositis, myasthenia gravis exacerbation, and myocarditis, that may co-occur and demand early recognition and aggressive immunosuppression [108]. A phase 2 trial study of pembrolizumab, showed grade ≥ 3 irAEs occurring in 71% of thymoma vs. 15% of thymic carcinoma, underscoring the need to present efficacy and toxicity separately by histology [109]. Pre-treatment evaluation should incorporate screening for paraneoplastic autoimmunity (clinical assessment for myasthenia gravis with appropriate work-up) as recommended in disease guidelines, and proactive biomarker monitoring (e.g., baseline and early on-treatment troponin and creatine kinase) as suggested by trial investigators and translational observations linking anti-Acetylcholine receptor (AChR) antibodies to muscle irAEs [110].

The efficacy data for anti-PD-1 and anti-PD-L1 will be discussed separately based on the respective studies and are summarized in Table 5.

### 10.1. Pembrolizumab

Giaccone et al. conducted a phase II trial in which pembrolizumab was administered to 40 patients with recurrent thymic carcinoma and reported an overall response rate (ORR) of 22.5%, with disease control achieved in 75% of the patients, a median response duration of almost 2 years, median progression-free survival (mPFS) of 4.2 months, and a median overall survival (mOS) of 24.9 months [111]. Data analysis showed that the majority of patients with higher PD-L1 expression had partial or complete response, and more than 80% of those with low or negative PD-L1 expression had progressive disease. There was a correlation between IFN-γ signature expression and response to pembrolizumab, and between TP53 mutations and reduced PD-L1 expression, which implied shorter overall survival. Severe immune-related adverse events, including myocarditis and polymyositis, were observed in 15% of the patients [111]. Another trial that evaluated pembrolizumab in this scenario was a phase II trial, which included recurrent thymic carcinoma and thymoma. The ORR was 19.2% for thymic carcinoma and 28.6% for thymomas, with a median duration of response of 9.7 months for thymic carcinoma and not reached for thymomas; mPFS was 6.1 months for both groups, while mOS was 14.5 months for thymic carcinoma and not reached for thymomas. More than 35% of patients with high PD-L1 expression achieved a partial response (PR), and none of the patients with negative PD-L1 expression responded to treatment. Immune-related adverse events were much more common in the thymoma group (71%) than in the thymic carcinoma group (15%) [119].

### 10.2. Nivolumab

Nivolumab was studied in the phase II single arm trial PRIMER, where 15 patients with recurrent or unresectable thymic carcinoma received immunotherapy alone. Although the disease control rate (DCR) was 73% and mOS was 14.1 months, with a manageable toxicity profile, the trial was stopped early owing to a lack of objective responses, with an mPFS of 3.8 months [112]. A retrospective study by Ak and Aydiner found a 66.7% objective response rate and 100% DCR in five evaluable patients with thymomas and thymic carcinomas [94]. More recently, a study of nivolumab plus ipilimumab (EORTC-ETOP NIVOTHYM) for advanced or relapsed type B3 thymoma or thymic carcinoma failed to show improvement in outcomes in the combination arm and had more adverse events, but the 40% PFS rate at 6 months was not reached [120].

### 10.3. Avelumab

Avelumab was tested in a phase I trial in patients with thymic carcinoma and thymoma, observing an objective response rate of nearly 30%. There was a higher incidence of immune-related adverse events compared to other adult malignancies treated with avelumab in previous trials, with 38% of patients experiencing grades 3 and 5 such as myositis and respiratory muscle insufficiency [114]. Further analysis revealed that patients who had some response had higher pretreatment absolute lymphocyte counts and lower percentages of B cells, regulatory T cells, and natural killer cells than non-responders. Immune infiltrates in the tumors showed a shift from immature to mature CD8+ T cells in responders. In addition, responders had greater T cell receptor diversity. Another study by the same group suggested that pre-existing anti-acetylcholine receptor autoantibodies and B-cell lymphopenia might increase the risk of developing myositis following avelumab treatment, even in the absence of a history of autoimmune disease, highlighting the need for further investigation of potential biomarkers [121].

The combination of immunotherapy and anti-angiogenic therapy also appears to be promising. Data from the CAVEATT trial, a phase II trial that analyzed the combination of avelumab and axitinib, showed an ORR above 30% and meaningful PFS (7.5 months), especially in the thymic carcinoma population [117].

### 10.4. Atezolizumab

Atezolizumab monotherapy was tested in a phase II basket trial with the primary endpoint of a prespecified non-progression rate (NPR). Even though the thymoma cohort had more than 76% NPR, this was not confirmed in the interim analyses [115]. A combination of atezolizumab plus carboplatin and paclitaxel was tested in 48 patients with thymic carcinoma in a phase II Japanese trial, with an ORR of 56%, with more than half of the patients achieving partial response [116].

### 10.5. KN046

This bispecific anti-PD-L1/PD1 and CTLA-4/CD80/CD86 antibody was tested in a phase II study that showed an ORR of approximately 16% [118]. Although it showed antitumor activity, another study was terminated because the data collected did not support the study endpoints [122].

Other clinical trials involving chemotherapy and immunotherapy are summarized in Table 6.

## 11. Future Directions

As new strategies have emerged, including immunotherapy, targeted therapy, and more recently, antibody-drug conjugates, we aim to enhance treatment response, improve tolerability, and extend the survival of these patients with thymic malignancies. Additionally, a better understanding of the optimal combination of these approaches, as well as the most effective sequencing of treatments, is crucial, as patients achieve longer survival. The thymus is also the richest source of extracellular vesicles among solid tissues and this heterogeneity has the potential for discovery of innovative new biomarkers that may lead to novel therapeutic strategies in the future. Specialized centers that manage a high number of cases play a key role in optimizing patient outcomes. Providing multimodal treatment is important, especially for non-metastatic diseases, and further research is needed to explore the potential benefits of maintenance therapy in this setting. Also, further improvement would be to focus on studies with endpoints based on FFR (freedom-from-recurrence) in early-stage tumors and TTP (time to progression) in advanced-stage disease, because survival outcomes are not the most appropriate measure for thymic malignancies according to International Thymic Malignancy Interest Group (ITMIG).

## 12. Conclusions

Thymic malignancies represent a rare and diverse group of tumors, with thymomas and thymic carcinomas being the most common subtypes. Thymomas are often associated with autoimmune disorders, such as myasthenia gravis, while thymic carcinomas rarely exhibit these phenomena. Despite advances in the understanding of their molecular biology and classification, prognosis remains largely dependent on staging. Surgical resection remains the primary treatment for early-stage tumors, and while chemotherapy has historically been the mainstay for advanced stages, recent innovations in immunotherapy and targeted treatments are expanding therapeutic options. Ongoing research into molecular subsets promises to further refine the treatment strategies for thymic malignancies.

## Figures and Tables

**Figure 1 cancers-17-03622-f001:**
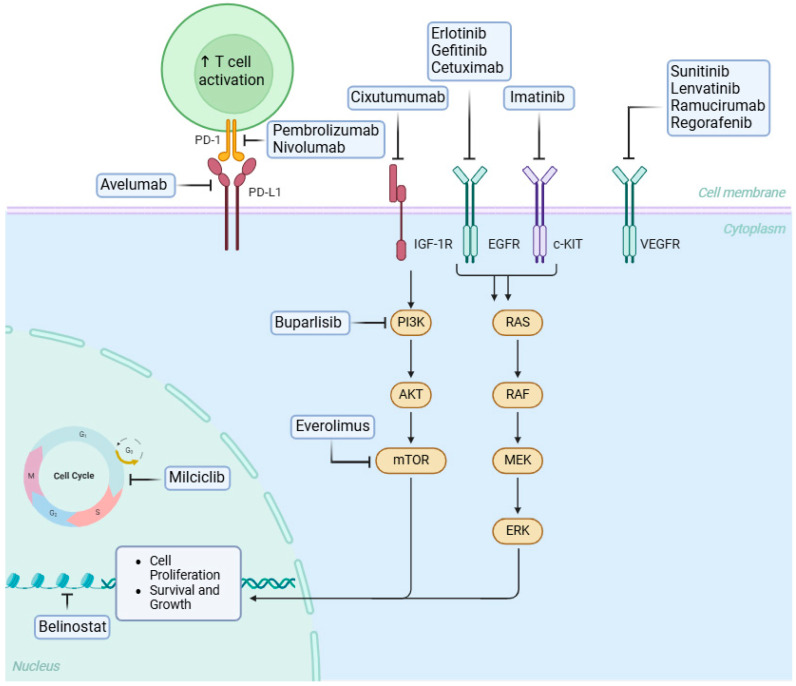
Mechanisms of action of targeted therapy and immunotherapy. The blunt head arrow denotes inhibition. Created in BioRender.

**Table 2 cancers-17-03622-t002:** (1) TNM 9th edition for Thymic Epithelial Tumors [32]. (2) Stage groups [32].

**(1)**					
**T**		**N**		**M**	
**T1**	**T1a: ≤5 cm**	**N0**	No regional nodal metastasis	**M0**	No pleural/pericardial/distant metastasis
	T1b: >5 cm				
**T2**	Direct invasion of pericardium and/or lung or phrenic nerve	**N1**	Anterior nodes	**M1**	M1a: Separate pleural or pericardial nodules (“implants”)
					M1b: Intraparenchymal lung nodule(s) or distant metastasis
**T3**	Invasion of brachiocephalic vein, SVC, chest wall, or extrapericardial pulmonary arteries/veins	**N2**	Deep intrathoracic or cervical nodes		
**T4**	Invasion of aorta or arch vessels, intrapericardial pulmonary arteries/veins, myocardium, trachea, or esophagus				
**(2)**					
**Stage**	**TNM**				
**I**	T1a–b N0 M0				
**II**	T2 N0 M0				
**IIIA**	T3 N0 M0				
**IIIB**	T4 N0 M0				
**IVA**	Any T N1 M0/Any T N0–N1 M1a				
**IVB**	Any T N2 M0–M1a/Any T Any N M1b				

**Table 4 cancers-17-03622-t004:** Clinical trials currently evaluating various targeted therapies for Thymomas or Thymic Carcinomas.

Trial	NCT	Drug	Status
Phase II Trial of Sacituzumab Govitecan in Patients with Advanced Thymic Epithelial Tumors [100]	06248515	Sacituzumab Govitecan	Recruiting
PT-112 in Subjects with Thymoma and Thymic Carcinoma [101]	05104736	PT-112, a first-in-class metallo-pyrophosphate conjugate	Recruiting
Selective TrkA Inhibitor VMD-928 to Treat TrkA Overexpression Driven Solid Tumors or Lymphoma [102]	03556228	Selective TrkA Inhibitor VMD-928	Recruiting
Study of Nanrilkefusp Alfa Alone and with Pembrolizumab in Adult Patients with Advanced/Metastatic Solid Tumors [103]	04234113	Nanrilkefusp	Active, not recruiting
A Study of Safety and Efficacy of KFA115 Alone and in Combo with Pembrolizumab in Patients with Select Advanced Cancers [104]	05544929	KFA115	Recruiting
Phase I/II Eval Safety & Prelim Activity Nivolumab Comb W/Vorolanib Pts W/Refractory Thoracic Tumors [105]	03583086	Vorolanib	Active, not recruiting
Bintrafusp Alfa (M7824) in Subjects with Thymoma and Thymic Carcinoma [106]	04417660	Bintrafusp Alfa (M7824)	Recruiting
Combination of Pembrolizumab and Lenvatinib, in Pre-treated Thymic Carcinoma patients (PECATI) [86]	04710628	Lenvatinib + Pembrolizumab	Active, not recruiting

**Table 5 cancers-17-03622-t005:** Clinical trial data of immune checkpoint inhibitors for recurrent thymic epithelial tumor.

Trial	Drug	Number of Patients	TET Histology	ORR (%)	Median PFS (Months)	Median OS (Months)
Giaccone, G et al. 2018 [111]	Pembrolizumab	40	Thymic carcinoma	22.5	4.2	24.9
Cho J, et al. 2019 [109]	Pembrolizumab	33	Thymoma Thymic carcinoma	28.6 19.2	6.1 6.1	NR 14.1
Katsuya Y et al. 2019 [112]	Nivolumab	15	Thymic carcinoma	0	3.8	14.1
Girard, N et al. 2023 [113]	Nivolumab	49	B3 Thymoma Thymic carcinoma	12	6	21.3
Rajan, A 2019 [114]	Avelumab	22	Thymoma Thymic carcinoma	12 10	6.4 14.7	NR NR
Tabernero, J 2022 [115]	Atezolizumab	13	Thymoma	38.5	11.7	NE
Shukuya, T 2025 [116]	Atezolizumab + carboplatin + paclitaxel	48	Thymic carcinoma	56	NE	NE
Conforti, F 2022 [117]	Avelumab + axitinib	27 3 2	Thymic carcinoma Thymoma Mixed	34.4	7.5	26
Fang, W 2023 [118]	KN046	46	Thymic carcinoma	16.3	5.6	NE

NE = not estimable. NR = Not reached.

**Table 6 cancers-17-03622-t006:** Clinical trials currently under investigation for chemotherapy and immunotherapy for Thymomas and Thymic Carcinomas.

Trial	NCT	Chemotherapy	Status
Adjuvant Treatment for Incomplete Resection Thymoma or Thymic Carcinoma [123]	02633514	Cisplatin and Etoposide	Recruiting
Ramucirumab and Carbo-Paclitaxel for Untreated Thymic Carcinoma/B3 Thymoma with Carcinoma [124]	03921671	Ramucirumab and Carbo-Paclitaxel	Active, not recruiting
Postoperative Adjuvant Chemotherapy for Thymic Cancer (FUSCC-Thymic 3) [125]	06402708	Medium dose of docetaxel, cisplatin, 5-FU	Recruiting
Carboplatin and Paclitaxel with or Without Ramucirumab in Treating Patients with Locally Advanced, Recurrent, or Metastatic Thymic Cancer That Cannot Be Removed by Surgery [126]	03694002	Carboplatin-paclitaxel with or without ramucirumab	Active, not recruiting
Chemotherapy Plus Cetuximab Followed by Surgical Resection in Patients with Locally Advanced or Recurrent Thymoma or Thymic Carcinoma [126]	01025089	Cetuximab, Cisplatin, Doxorubicin, Cyclophosphamide	Active, not recruiting
First-line CBDCA/PTX/LEN/Pembrolizumab Combination for Previously Untreated Advanced or Recurrent Thymic Carcinomas (Artemis) [127]	05832827	Carboplatin/ Paclitaxel/ Lenvatinib/ Pembrolizumab combination	Recruiting
Pembrolizumab and Sunitinib Malate in Treating Participants with Refractory Metastatic or Unresectable Thymic Cancer [128]	03463460	Pembrolizumab and Sunitinib	Recruiting
Pembrolizumab in Treating Participants with Unresectable Thymoma or Thymic Cancer [129]	03295227	Pembrolizumab	Recruiting

## Data Availability

Not applicable.

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
