# Peer review of "Current Clinical Paradigm and Therapeutic Advancements in Thymic Malignancies: A Narrative Review"

_cancers, 2025, doi:10.3390/cancers17223622_

Round 1

Reviewer 1 Report (Previous Reviewer 4)

Comments and Suggestions for Authors

1. In addition to TABLE 1, please supplement the table of thymoma subtypes in the WHO classification. 
2. Figure 1 is not clear. 
In Tables 3, 4 and 5, the TET pathological types should be marked as in Table 4. 
4. There are spelling mistakes in Table 4. Please correct them.

Author Response

Reviewer 1

  1. In addition to TABLE 1, please supplement the table of thymoma subtypes in the WHO classification. 
    A supplementary table has been added for the thymoma subtypes.
    Figure 1 is not clear. 
    There is a raw image file provided for Figure 1 with the submission. The low resolution Figure 1 in the PDF version is due to Cancers formatting which we cannot modify. We will ensure that the final proof uses the high quality figure file.
    3. In Tables 3, 4 and 5, the TET pathological types should be marked as in Table 4.

Tables 3, 4, and 5 represent different figures. TET histology was included in Table 5 as the results of the trials are histology specific. Table 3 is drug specific results table rather than a histology specific table. Table 4 is ongoing trials with wide enrollment criteria, including solid tumors, and TET histology would overwhelm the scope of the table.   
4. There are spelling mistakes in Table 4. Please correct them.
The typos have been corrected.

Reviewer 2 Report (New Reviewer)

Comments and Suggestions for Authors

General Assessment:

The manuscript presents a well written review summarizing the state of the art on Thymoma disease. The authors are recognized researchers from prestigious cancer centers. However, some revisions are suggested before publication.

Major points: Text Structure:

Reformat the text to reduce the number of sections.

Simplify the language to improve readability.

Content Addition:

In Section 2 (Incidence and Epidemiology), where the correlation between thymoma and myasthenia gravis is mentioned, add a short section discussing the diagnostic and prognostic relevance of inflammatory and metabolic biomarkers for patient stratification.

Minor Points

  • Remove yellow underlining throughout the manuscript.
  • Rename the section “Systemic therapy today” to “Current systemic therapy.”
  • Merge Sections 5 and 6 under the new title “Genomic landscape.”

Author Response

Reviewer 2

General Assessment:

The manuscript presents a well written review summarizing the state of the art on Thymoma disease. The authors are recognized researchers from prestigious cancer centers. However, some revisions are suggested before publication.

Major points: Text Structure:

Reformat the text to reduce the number of sections.

Simplify the language to improve readability.

Thank you for this feedback and we have restructured some of the manuscript as well as reviewed the manuscript text.

Content Addition:

In Section 2 (Incidence and Epidemiology), where the correlation between thymoma and myasthenia gravis is mentioned, add a short section discussing the diagnostic and prognostic relevance of inflammatory and metabolic biomarkers for patient stratification.

It is still yet to be determined which predictive biomarkers are relevant to paraneoplastic syndrome that are related to thymomas. Further evaluation is needed in the future. 

Minor Points

  • Remove yellow underlining throughout the manuscript.
    The yellow was track changes as requested by the editor.
  • Rename the section “Systemic therapy today” to “Current systemic therapy.”

This has been renamed.

  • Merge Sections 5 and 6 under the new title “Genomic landscape.”

This has been merged.

Reviewer 3 Report (New Reviewer)

Comments and Suggestions for Authors

The authors did a great job in summarizing historical and modern, clinical and molecular, previous and recent datasets available in the context of thymic tumors. This is not an easy task due to the relative rarity of these tumors. A real bonus is the extended view on the currently available immune therapy options. Only minor criticism may apply to this review article: 1) a few sentences would benefit from rephrasing (e.g. the very first sentence of the abstract contains two verbs: ‘are arise’, please resolve this and other such scarce issues), 2) future directions could also be extended (e.g. the thymus is the richest source of extracellular vesicles among solid tissues, potentially providing biomarkers and novel intervention strategies etc).

Author Response

Reviewer 3

The authors did a great job in summarizing historical and modern, clinical and molecular, previous and recent datasets available in the context of thymic tumors. This is not an easy task due to the relative rarity of these tumors. A real bonus is the extended view on the currently available immune therapy options. Only minor criticism may apply to this review article:
1) a few sentences would benefit from rephrasing (e.g. the very first sentence of the abstract contains two verbs: ‘are arise’, please resolve this and other such scarce issues)
We would like to thank the reviewer for their feedback and have corrected this typo and have reviewed the manuscript for other similar typos.
, 2) future directions could also be extended (e.g. the thymus is the richest source of extracellular vesicles among solid tissues, potentially providing biomarkers and novel intervention strategies etc).

We have expanded the future directions section as requested.

This manuscript is a resubmission of an earlier submission. The following is a list of the peer review reports and author responses from that submission.

Round 1

Reviewer 1 Report

Comments and Suggestions for Authors

This is a timely and excellent summary of the biology and treatment of thymic epithelial tumors. The narrative review is multidisciplinary, but has a particular focus on systemic therapies--the purview of the medical oncologist. In this regard, the review summarizes several recent advances, including anti-PD-1 combination strategies (noted under each section with individual agents). The manuscript can be strengthened through description of the findings of Proto and colleagues that ramucirumab improves responses to carbo/taxol (now published, DOI: 10.1016/j.annonc.2024.06.002). I would also report the findings of the recent update of the NIVOTHYM trial in which nivo/ipi did not demonstrate efficacy in B3 thymoma/thymic carcinoma (https://doi.org/10.1200/JCO.2025.43.16_suppl.8016). Otherwise, the manuscript is acceptable for publication.

Author Response

This is a timely and excellent summary of the biology and treatment of thymic epithelial tumors. The narrative review is multidisciplinary, but has a particular focus on systemic therapies--the purview of the medical oncologist. In this regard, the review summarizes several recent advances, including anti-PD-1 combination strategies (noted under each section with individual agents). The manuscript can be strengthened through description of the findings of Proto and colleagues that ramucirumab improves responses to carbo/taxol (now published, DOI: 10.1016/j.annonc.2024.06.002). I would also report the findings of the recent update of the NIVOTHYM trial in which nivo/ipi did not demonstrate efficacy in B3 thymoma/thymic carcinoma (https://doi.org/10.1200/JCO.2025.43.16_suppl.8016). Otherwise, the manuscript is acceptable for publication.
Response: We thank the reviewer for this valuable observation, we have expanded on these studies within the manuscript and added these references: 1) DOI: 10.1016/j.annonc.2024.06.002 and 2) doi.org/10.1200/JCO.2025.43.16_suppl.8016

Reviewer 2 Report

Comments and Suggestions for Authors

The authors did a comprehensive review on thymic tumors. The manuscript reads more like a textbook chapter than a review paper. And there are some major problems, showing that the authors are actually not very familiar with the disease or recent advances in its diagnosis and management.

Incidence and Epidemiology: The authors stated that ‘Nevertheless, they are responsible for up to 50% - 63% of anterior mediastinal masses’. This might be the incidence in surgical series. Please refer to the ITMIG global imaging database study for accurate figures.

Histological classification: Thymic carcinomas have been separated from thymomas since the 2015 WHO classification. It is no longer named Type C.

Staging: The MK staging is obsolete and should have been abandoned since the 8th TNM staging system. The 9th TNM staging was published in 2023 and has been in effect since Jan. 2025.

Systemic therapy today: ‘In the neoadjuvant setting, a few retrospective studies showed no difference between upfront approach versus neoadjuvant treatment in regard to OS, PFS, pathological complete response or R0 resections 72’.This is a wrong reference (Cho, J., Kim, H.S., Ku, B.M., Choi, Y.L., Cristescu, R., Han, J., Sun, J.M., Lee, S.H., Ahn, J.S., Park, K., and Ahn, M.J. (2019). Pembrolizumab for Patients With Refractory or Relapsed Thymic Epithelial Tumor: An Open-Label Phase II Trial. J Clin Oncol 37, 2162-2170. 10.1200/jco.2017.77.3184), which is actually on second-line immunotherapy for advanced/recurrent thymic tumors.in addition, hyperthermic intrathoracic chemotherapy (HITHOC)is a local irrigation with chemotherapy agents rather than systemic chemotherapy.

Radiotherapy:’Postoperative radiation therapy is usually indicated for staging above stage IIB or positive surgical margins, with one trial showing OS benefit for stage III and IV’. Actually studies using large global, regional, national database showed conflicting results. Many of those studies (the JART, ChART, etc) are not even cited or discussed here.

Targeted Therapy Developments and Novel Immunotherapy Approaches: The authors talked a lot about endpoints using survivals. But thymic malignancies, esp. thymomas are indolent tumors. Survival outcomes are not ideal endpoints for the study of these tumors. ITMIG recommends FFR for early-stage tumors and TTP for advanced diseases.

Author Response

Reviewer 2
Comments and Suggestions for Authors
The authors did a comprehensive review on thymic tumors. The manuscript reads more like a textbook chapter than a review paper. And there are some major problems, showing that the authors are actually not very familiar with the disease or recent advances in its diagnosis and management.
Incidence and Epidemiology: The authors stated that ‘Nevertheless, they are responsible for up to 50% - 63% of anterior mediastinal masses’. This might be the incidence in surgical series. Please refer to the ITMIG global imaging database study for accurate figures.
Response: We have included language in the manuscript on this and the ITMIG reference.
Histological classification: Thymic carcinomas have been separated from thymomas since the 2015 WHO classification. It is no longer named Type C.
Response: We thank the reviewer for this valuable observation, we have removed table 1 that addressed the tumor as type C.
Staging: The MK staging is obsolete and should have been abandoned since the 8th TNM staging system. The 9th TNM staging was published in 2023 and has been in effect since Jan. 2025.
Response: We thank the reviewer for this valuable observation, we previously kept MK staging for historical purposes, but we have added the 9th TNM staging.
Systemic therapy today: ‘In the neoadjuvant setting, a few retrospective studies showed no difference between upfront approach versus neoadjuvant treatment in regard to OS, PFS, pathological complete response or R0 resections 72’.This is a wrong reference (Cho, J., Kim, H.S., Ku, B.M., Choi, Y.L., Cristescu, R., Han, J., Sun, J.M., Lee, S.H., Ahn, J.S., Park, K., and Ahn, M.J. (2019). Pembrolizumab for Patients With Refractory or Relapsed Thymic Epithelial Tumor: An Open-Label Phase II Trial. J Clin Oncol 37, 2162-2170. 10.1200/jco.2017.77.3184), which is actually on second-line immunotherapy for advanced/recurrent thymic tumors.in addition, hyperthermic intrathoracic chemotherapy (HITHOC)is a local irrigation with chemotherapy agents rather than systemic chemotherapy.
Response: We thank the reviewer for this valuable observation, we have corrected the reference and moved HITHOC to a different paragraph, other than systemic therapy
Radiotherapy:’Postoperative radiation therapy is usually indicated for staging above stage IIB or positive surgical margins, with one trial showing OS benefit for stage III and IV’. Actually studies using large global, regional, national database showed conflicting results. Many of those studies (the JART, ChART, etc) are not even cited or discussed here.
Response: We thank the reviewer for this valuable observation, we have added additional text in the manuscript and other references.
Targeted Therapy Developments and Novel Immunotherapy Approaches: The authors talked a lot about endpoints using survivals. But thymic malignancies, esp. thymomas are indolent tumors. Survival outcomes are not ideal endpoints for the study of these tumors. ITMIG recommends FFR for early-stage tumors and TTP for advanced diseases.
Response: We thank the reviewer for this valuable observation, we have addressed this critique for ideal endpoints in the future directions section.

Reviewer 3 Report

Comments and Suggestions for Authors

Dear Editor,

I was pleased to review the revised form of the manuscript entitled ''Current clinical paradigm and therapeutic advancements in thymic malignancies: a narrative review''. The authors reviewed the histology and molecular subtypes of thymic epithelial tumors and examined therapeutic approaches, including current standard treatment regimens and drugs currently in clinical trials. The review is well-prepared and effectively communicates treatment selection and outcomes. I congratulate the authors on this valuable study for readers.

Sincerely

Author Response

Reviewer 3
Comments and Suggestions for Authors
Dear Editor,
I was pleased to review the revised form of the manuscript entitled ''Current clinical paradigm and therapeutic advancements in thymic malignancies: a narrative review''. The authors reviewed the histology and molecular subtypes of thymic epithelial tumors and examined therapeutic approaches, including current standard treatment regimens and drugs currently in clinical trials. The review is well-prepared and effectively communicates treatment selection and outcomes. I congratulate the authors on this valuable study for readers.
Response: We thank the reviewer for their kind feedback and agree on the timeliness of this review.

Reviewer 4 Report

Comments and Suggestions for Authors

1, The text mentioned that the WHO pathological classification system proposed the classification of types A, AB, B1, B2, B3, and C in 2021. In fact, this classification method can be traced back to the 2004 edition of the WHO classification. Please verify the time and make the necessary corrections.

2, According to the current consensus, the classification concept of "benign" for thymic tumors is no longer used. It is suggested that the change and the latest literature it is based on be supplemented in the pathological classification section to reflect the progress in medical understanding.

3. Although the article systematically reviews various thymoma classification systems since the 1960s, it lacks an analysis and comparison of their practical application value in clinical decision-making (such as surgical methods and adjuvant treatment options) and prognosis judgment. It is suggested that the discussion on the guiding significance of different classification systems for the formulation of treatment plans be added.

4. Organize the research and results related to targeted therapy in the article into a table to make them visually presentable.

5. Figure 1 The image is not clear. It is recommended to provide a higher resolution image.

6. The application of radiotherapy should be elaborated in a hierarchical manner.
It is suggested that the authors further clarify the specific indications and effects of radiotherapy in different stages and pathological subtypes, so as to make the content more instructive.

7. Table 4 has inconsistent font styles.

8. The chapter on immunotherapy should be discussed in a refined and layered manner.
Currently, the section on immunotherapy mainly consists of a simple listing of research results. Suggestion: The research data on thymoma and thymic carcinoma should be separately organized and discussed, as their biological characteristics and immunotoxic risks are significantly different. The system summarizes all the immune-related adverse events (irAEs) that occurred in each study, such as myositis, myocarditis, endocrine abnormalities, etc., and makes necessary classifications and risk warnings.

Author Response

Reviewer 4
Comments and Suggestions for Authors
1, The text mentioned that the WHO pathological classification system proposed the classification of types A, AB, B1, B2, B3, and C in 2021. In fact, this classification method can be traced back to the 2004 edition of the WHO classification. Please verify the time and make the necessary corrections.
Response: Thank you for pointing this out. We have corrected this in the manuscript.
2, According to the current consensus, the classification concept of "benign" for thymic tumors is no longer used. It is suggested that the change and the latest literature it is based on be supplemented in the pathological classification section to reflect the progress in medical understanding.
Response: We thank the reviewer for this valuable observation. We have revised the pathological classification section accordingly and supplemented it with the most recent literature, including the 2021 WHO Classification of Tumors of the Thymus and Mediastinum.
3. Although the article systematically reviews various thymoma classification systems since the 1960s, it lacks an analysis and comparison of their practical application value in clinical decision-making (such as surgical methods and adjuvant treatment options) and prognosis judgment. It is suggested that the discussion on the guiding significance of different classification systems for the formulation of treatment plans be added.
Response: We thank the reviewer for this valuable suggestion. In response, we have expanded the section on the WHO histological classification to emphasize its practical clinical relevance.
4. Organize the research and results related to targeted therapy in the article into a table to make them visually presentable.
Response: In the revised version of the manuscript, we have organized the research and clinical results regarding targeted therapies for thymic malignancies into a comprehensive table. This table summarizes the therapeutic classes, specific agents, mechanisms of action, key clinical trial outcomes.
5. Figure 1 The image is not clear. It is recommended to provide a higher resolution image.
Response: We have added a higher resolution image and the original figure file was included.
6. The application of radiotherapy should be elaborated in a hierarchical manner. It is suggested that the authors further clarify the specific indications and effects of radiotherapy in different stages and pathological subtypes, so as to make the content more instructive.
Response: The section on radiotherapy has been expanded and reorganized in a hierarchical manner. We now clarify the specific indications and effects of radiotherapy according to disease stage and pathological subtypes, highlighting its role in early, intermediate, and advanced settings.
7. Table 4 has inconsistent font styles.
Response: Thank you for pointing it out, we have corrected font style for table 4.
8. The chapter on immunotherapy should be discussed in a refined and layered manner. Currently, the section on immunotherapy mainly consists of a simple listing of research results. Suggestion: The research data on thymoma and thymic carcinoma should be separately organized and discussed, as their biological characteristics and immunotoxic risks are significantly different. The system summarizes all the immune-related adverse events (irAEs) that occurred in each study, such as myositis, myocarditis, endocrine abnormalities, etc., and makes necessary classifications and risk warnings.
Response: We have revised the section to discuss thymoma and thymic carcinoma separately marking the difference between them regarding immune-related adverse events.
